# Randomised controlled trial to determine the efficacy and safety of prescribed water intake to prevent kidney failure due to autosomal dominant polycystic kidney disease (PREVENT-ADPKD)

Annette T Y Wong,[1,2] Carly Mannix,[1,2] Jared J Grantham,[3] Margaret Allman-Farinelli,[4] Sunil V Badve,[5] Neil Boudville,[6] Karen Byth,[7] Jessie Chan,[8] Susan Coulshed,[9] Marie E Edwards,[10] Bradley J Erickson,[10] Mangalee Fernando,[11] Sheryl Foster,[12,13] Imad Haloob,[14] David C H Harris,[1,2] Carmel M Hawley,[15] Julie Hill,[8] Kirsten Howard,[16] Martin Howell,[16] Simon H Jiang,[17,18] David W Johnson,[15] Timothy L Kline,[10] Karthik Kumar,[19] Vincent W Lee,[1,2,20] Maureen Lonergan,[21] Jun Mai,[22] Philip McCloud,[8] Anthony Peduto,[12] Anna Rangan,[4] Simon D Roger,[23] Kamal Sud,[2,24,25] Vincent Torres,[10] Eswari Vliayuri,[26] Gopala K Rangan[1,2]

For numbered affiliations see end of article.

**Correspondence to**
Dr Gopala K Rangan;
g.rangan@sydney.edu.au

## ABSTRACT

**Introduction** Maintaining fluid intake sufficient to reduce arginine vasopressin (AVP) secretion has been hypothesised to slow kidney cyst growth in autosomal dominant polycystic kidney disease (ADPKD). However, evidence to support this as a clinical practice recommendation is of poor quality. The aim of the present study is to determine the long-term efficacy and safety of prescribed water intake to prevent the progression of height-adjusted total kidney volume (ht-TKV) in patients with chronic kidney disease (stages 1–3) due to ADPKD.

**Methods and analysis** A multicentre, prospective, parallel-group, open-label, randomised controlled trial will be conducted. Patients with ADPKD (n=180; age ≤65 years, estimated glomerular filtration rate (eGFR) ≥30 mL/min/1.73 m$^2$) will be randomised (1:1) to either the control (standard treatment+usual fluid intake) or intervention (standard treatment+prescribed fluid intake) group. Participants in the intervention arm will be prescribed an individualised daily fluid intake to reduce urine osmolality to ≤270 mOsmol/kg, and supported with structured clinic and telephonic dietetic review, self-monitoring of urine-specific gravity, short message service text reminders and internet-based tools. All participants will have 6-monthly follow-up visits, and ht-TKV will be measured by MRI at 0, 18 and 36 months. The primary end point is the annual rate of change in ht-TKV as determined by serial renal MRI in control vs intervention groups, from baseline to 3 years. The secondary end points are differences between the two groups in systemic AVP activity, renal disease (eGFR, blood pressure, renal pain), patient adherence, acceptability and safety.

**Ethics and dissemination** The trial was approved by the Human Research Ethics Committee, Western Sydney Local Health District. The results will inform clinicians, patients and

### Strengths and limitations of this study

► A major strength of the study is that it has a randomised controlled trial design and will provide high-level evidence regarding the long-term efficacy of prescribed water intake on the progression of autosomal dominant polycystic kidney disease.
► Other strengths include the duration of follow-up (3 years) and the use of height-adjusted total kidney volume as the primary outcome measure.
► Lastly, the study intervention will be implemented using a multipronged approach using self-monitoring, dietetic intervention and mobile phone technology.
► The limitations of the study are that the trial intervention is unblinded and is reliant on the behavioural change to drinking habits of the participant.

policy-makers regarding the long-term safety, efficacy and feasibility of prescribed fluid intake as an approach to reduce kidney cyst growth in patients with ADPKD.

**Trial registration number** ANZCTR12614001216606.

## INTRODUCTION

Autosomal dominant polycystic kidney disease (ADPKD) is the most common genetic kidney disease in adults, affecting one in every 2500 individuals, and the cause of kidney failure in 5%–10% of the dialysis population worldwide.[1] It is due to heterozygous germline mutations in *PKD1* (85%) or *PKD2* (15%), which encode the

transmembrane protein polycystin-1 and calcium ion channel polycystin-2, respectively.[1] These proteins maintain the differentiated structure of the nephron during health and disease.[1] The clinical hallmark of ADPKD is the presence of numerous nephron-derived cysts in the kidney, which form in early childhood and grow by 5%–10% per year, such that by midlife the kidney is about five times larger than normal (1.0 vs 0.2 kg),[2] causing chronic pain and hypertension. The expanding cysts also compress healthy kidney tissue, leading to progressive chronic kidney disease (CKD) and renal replacement therapy in ~50% of affected people by age of 60 years.[3] Currently, there is no 'cure' for ADPKD, and the ideal therapy to stop kidney cyst growth and prevent end-stage kidney disease (ESKD) will be one with few side effects, as it will need to be taken lifelong.[4]

Arginine vasopressin (AVP) is a posterior pituitary hormone with a recognised physiological role in maintaining water homeostasis.[5] It is released in response to hypovolaemia and hyperosmolality, and binds to $V_2$ receptors on the principal cells of the collecting duct in the kidney, causing reabsorption of water from the tubular lumen.[5] Renal cysts are derived from the principal cells of the collecting duct of the nephron.[6,7] However, the epithelial cells lining the cysts respond abnormally to AVP by activating intracellular cyclic adenosine monophosphate signalling, which stimulates proliferation and luminal fluid secretion, causing cyst growth. In rats, the congenital deficiency of AVP completely abrogated renal cyst formation and growth,[8] providing compelling evidence that AVP has a critical role in cystogenesis and that its inhibition at an early stage of disease could markedly reduce the risk of developing ESKD in ADPKD. In this regard, small-molecule vasopressin-receptor antagonists have been shown to be highly effective in reducing cyst growth in preclinical studies,[9] and in humans, a randomised controlled trial showed that 3 years of treatment with tolvaptan (a highly specific vasopressin-receptor antagonist) in early stages of ADPKD reduced the rate of increase in total kidney volume (TKV) by 50%, attenuated the decline in estimated glomerular filtration rate (eGFR) by 30% and reduced chronic kidney pain.[10]

For several years, it has been suggested that the suppression of AVP by increasing fluid intake could also slow renal cyst growth in ADPKD.[7,11] In support of the hypothesis, preclinical experiments in the *pck* rat model of PKD showed that increased water intake reduced kidney enlargement,[12,13] and comparison with separate studies imply that the efficacy might be similar (but with physiological differences) to vasopressin receptor antagonists.[14] However, whether this hypothesis is also true in humans with ADPKD remains unknown. The data available are limited to a single post hoc analytical study,[15] two short-term interventional trials (<1 week in duration) without control groups,[16,17] and a single, small, quasi-randomised observational cohort study of 12 months' duration, which paradoxically suggested

that increased fluid intake increases renal cyst growth.[18] Consequently, evidence-based clinical care guidelines have not included recommendations to increase fluid intake in patients with ADPKD and the matter remains controversial in clinical practice.[19] Consistent with this view, patients with ADPKD attending a consumer workshop also stated that the role of fluid intake was an ambiguous area that needed urgent prioritisation in clinical research.[20] An illustrative comment made by a workshop participant was: *'there needs to be consistency of what doctors say about drinking less or drinking more'*.[20]

Two recent clinical studies reported that prescribed fluid intake could be achieved over a period of 2–4 weeks in patients with ADPKD.[21,22] In addition, two randomised controlled trials are presently underway to address the role of fluid intake in ADPKD and CKD over a longer duration.[23,24] However, neither of these studies will specifically address the long-term efficacy of fluid intake on renal cyst growth in ADPKD. Hence, the aim of the current study is to determine the efficacy and safety of prescribed water intake to *prevent* the progression of TKV in CKD (stages 1–3) due to ADPKD (PREVENT-ADPKD) over a 3-year period. The trial commenced study activity in 2015 and as of July 2017 75% of the planned target recruitment has been attained. The current paper provides a summary of the clinical trial protocol.

## METHODS AND ANALYSIS
### Participants, design and registration
This is a prospective, parallel-group, open-label, multicentre randomised controlled trial, which will enrol 180 participants that meet the inclusion criteria (table 1). The planned recruitment period is up to 1.8 years, and began at Westmead Hospital in December 2015. For participants, the duration of the trial is 3.2 years, including the screening visit and the run-in, treatment and post-treatment periods (figure 1). The trial is registered (Australian New Zealand Clinical Trials Registry number 1261-4001-216-606) and will be conducted according to Good Clinical Practice and reported using Consolidated Standards of Reporting Trials guidelines.[25]

### Recruitment
Multiple strategies will be used to facilitate recruitment.[26,27] Participants will be identified from nephrologists practising at the study centres (see below), either through direct referral to the study team (email or verbal communication) or review of clinic letters and (if available) local databases. This approach will be supplemented with presentations at the study centres and adjacent hospitals (such as Royal North Shore, Concord and Royal Prince Alfred Hospitals in Sydney). In addition, participants will also be recruited passively through the internet, digital and print media advertising (listing on the websites of the PKD Foundation of Australia, the Australasian Kidney Trials Network,

| Table 1 | Inclusion and exclusion criteria |
| --- | --- |
| **Inclusion criteria** | |
| 1. | Adult patients providing informed consent, aged 18–65 years of age |
| 2. | Diagnosis of ADPKD, such as meeting the Pei-Ravine criteria[52] |
| 3. | eGFR (CKD-EPI) ≥30 mL/min/1.73 m² within 6 weeks of randomisation |
| **Exclusion criteria** | |
| 1. | Safety risk, eg, serum Na⁺ <135 mmol/L; requirement for medications with high risk of precipitating hyponatraemia, such as chronic use of diuretics; medical conditions that require fluid restriction, such as heart failure, chronic liver disease, nephrotic syndrome or generalised oedema; abnormalities in the voiding mechanism; pregnant or breastfeeding women |
| 2. | Contraindication to or interference with MRI assessments (eg, ferromagnetic prostheses, aneurysm clips, severe claustrophobia or other contraindications) |
| 3. | Risk of non-compliance with trial procedures (eg, history of non-compliance with medical therapy; history of substance abuse within the previous 2 years and/or participants who do not complete the required screening tests (24 hours urine, blood tests and baseline MRI) within 12 weeks of the screening visit) |
| 4. | Concomitant conditions or treatments likely to confound end point assessments (eg, poorly controlled diabetes, other known causes of CKD, renal cancer, single kidney or severe comorbid illnesses) |
| 5. | Participation in other clinical trials to slow ADPKD or CKD |
| 6. | TKV Mayo Clinic subclass 1A on screening (low risk of progression)[53] |

ADPKD, autosomal dominant polycystic kidney disease; CKD, chronic kidney disease; CKD-EPI, Chronic Kidney Disease Epidemiology Collaboration equation; eGFR, estimated glomerular filtration rate; TKV, total kidney volume.

Clinical Trials Connect and the University of Sydney; letters to Australian nephrologists, news items in the e-bulletins of Australian and New Zealand Society of Nephrology; newspaper advertisements; flyers placed in Renal Clinic waiting rooms in Sydney Hospitals). All identified and interested participants will be discussed with the treating nephrologist for their suitability and prescreened by telephone to tentatively determine their eligibility and verified using previous imaging and eGFR reports, prior to arranging their study visit.

### Study centres

The study centres will consist of a combination of University Teaching Hospitals, Medical Research Institutes and Private Consulting Rooms to facilitate the participants' ability to be involved in the trial. The centres will be

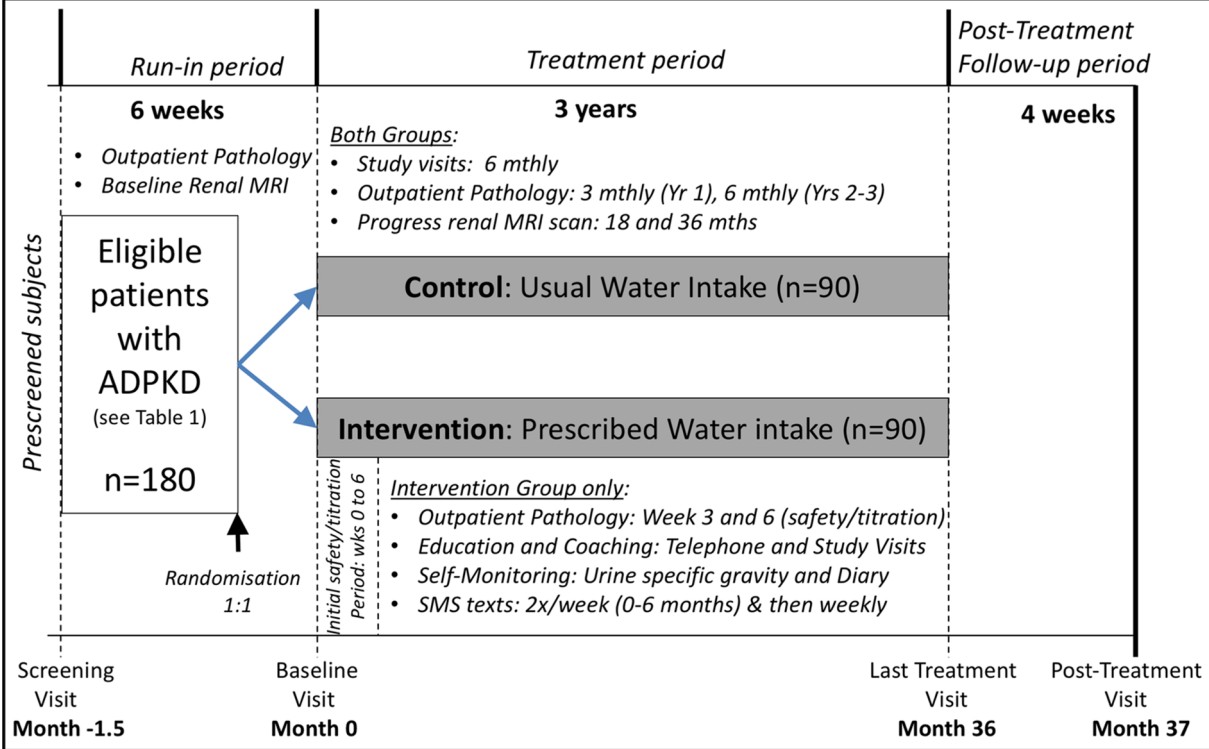

**Figure 1** Schema of the PREVENT-ADPKD trial design. Adapted from Torres et al.[10]

in Sydney (Westmead Institute for Medical Research, Westmead Hospital, Nepean Hospital, Norwest Private Hospital, Liverpool Hospital, Prince of Wales Hospital, St George Hospital, Mater Private Rooms), NSW Central Coast (Gosford Renal Research, Gosford Nephrology), Newcastle (John Hunter Hospital), Wollongong (Wollongong Hospital), Perth (Sir Charles Gairdner Hospital and the Harry Perkins Institute of Medical Research), Brisbane (Princess Alexandra Hospital) and Canberra (Canberra Hospital).

## Mobile study team

To enhance the efficiency of the trial and minimise its impact on the local resources of the study centres, a mobile study team[28] (based at the Westmead Institute for Medical Research and Westmead Hospital) consisting of research dietitians and a nephrologist, will visit the study centres to conduct research activity. The mobile study team will see several participants on the designated day of the visit, and be supported by local clinical research staff by provision of space and undertaking minor procedures such as blood collection.

## Study visits

1. *Screening visit (up to month 1.5).* If a patient meets all inclusion criteria and no exclusion criteria, he or she will be enrolled in the trial. At this visit, participants will have their medical and ADPKD history reviewed, usual fluid intake and kidney pain (using the Halt Progression of Polycystic Kidney Disease (HALT-PKD) study group questionnaire)[29] assessed. A venous blood sample and spot urine sample will also be collected for DNA analysis and biomarker assessment (see below). Due to the cost and time required to perform genetic testing, the effect of *PKD* mutation type on trial outcomes will be assessed retrospectively. Participants will then be asked to have two 24 hours urine collections and blood tests (for testing baseline electrolytes, eGFR and osmolality) as an outpatient at a local pathology collection centre, and a renal MRI (to assess baseline TKV) will be performed at an external radiology facility. The period from screening to randomisation visits (up to 12 weeks) will serve as the run-in period to confirm the participant's willingness to adhere to study procedures.
2. *Randomisation and baseline visit (month 0).* Follow-up medical and ADPKD histories will be taken and venous blood and urine samples will be collected at this visit. Participants randomised to the control group will continue with their usual (ad libitum) water intake and standard treatment. Participants in the intervention group will be advised to adjust their daily water intake for the next 36 months, in addition to continuation of their standard treatment, and be provided with specific instructions, described in the section under *study intervention (group B).*
3. *Treatment period.* The 3-year treatment period includes repeat outpatient blood and 24 hours urine collections (at 3-monthly intervals in year 1 and 6-monthly in years 2–3), progress MRI scans (at 18 and 36 months) and visits to the study centre every 6 months.
4. *Follow-up study visits (months 6, 12, 18, 24, 30 and 36).* Progress medical and ADPKD histories and further collections of venous blood and urine will be undertaken at follow-up study visits. Study staff will record answers to specific questions on adverse events (AEs) and kidney pain (using the HALT-PKD questionnaire). In both groups, quality of life will be assessed 6-monthly (month 0, 6, 12, 18, 24, 30 and 36) and usual fluid intake will be assessed annually (month 0, 12, 24 and 36). At the final treatment visit at month 36, intervention group participants will be advised that they may return to their previous ad libitum water intake.
5. *Post-treatment study visit.* This will occur at month 37 for all participants.

## Randomisation procedure

Participants will be randomly allocated in a 1:1 ratio (in permutated blocks of 4) to the control group or the intervention group, stratified by baseline eGFR (<60 or ≥60 mL/min/1.73 m$^2$). Randomisation and concealed allocation will be performed with a secure, web-based randomisation service (Randomize.net). The trial statisticians have generated a validated randomisation list.

## Study intervention (group B)

At the randomisation/baseline visit (month 0, see figure 1), the study dietitian will implement the following:
1. *Calculation of fluid prescription.* Participants will be advised to drink a prescribed volume of fluid per day (preferably tap water), based on the free water clearance formula, to reduce their urine osmolality to ≤270 mOsmol/L plus an amount to account for insensible losses (appropriate for climate and daily activity).[30 31] The calculation is as follows:[17]

$$\text{Prescribed fluid intake (mL)} = \frac{\text{total solutes (mOsmol) (mL)}}{270} + \text{insensible losses (mL)}$$

*Total solutes (urine osmolality×urine vol (mL)) is the mean derived from two 24 hours urine samples collected between the screening visit and the randomisation/baseline visit.*

2. *Dietary counselling.* A high dietary solute load (due to high salt and protein intakes) requires a higher fluid intake to maintain urine dilution.[11] Participants will be educated about the importance of dietary solute intake in determining obligatory urine volume (ie, the minimum urine volume required to excrete the daily solute load).[32] The study dietitian will take a detailed diet history and provide tailored dietary advice to enable participants to achieve and maintain a moderate protein intake (0.75–1.0 g/kg/day) and limit sodium intake to 80–100 mmol/day.[19] If the calculated prescription is >3 L/day, the participant will be advised to gradually increase fluid intake until target urine osmolality is reached and focus on

reducing dietary solutes to reduce the risk of hyponatraemia.

3. *Review of lifestyle and environmental factors.* The participant's lifestyle, personal preferences and occupation will be recorded and reviewed when providing individualised techniques for promoting adherence to fluid intake. Participants will be provided with 3×1 L reusable water bottles to help keep track of their fluid intake, and will be encouraged to drink evenly throughout the day and replenish with each episode of nocturia.

4. *Self-monitoring of fluid intake and treatment efficacy.* Aids have been developed to assist participants with self-monitoring, including a paper-based diary or web-based/smartphone compatible tool to self-monitor fluid intake, and urine dipsticks will be provided to record urine-specific gravity (USG). Participants will be shown how to read their USG and will be asked to test it during the late afternoon (16:00–20:00 hours) at least once daily in the first 2 weeks of the study, at least twice weekly for the first 6 months and then at least monthly for the duration of the study. An USG of ≤1.010 indicates a spot urine osmolality of ≤270 mmol/L, meaning that fluid intake in the past few hours has been adequate. Participants will also be briefed on receiving and responding to short message service (SMS) text messages (or emails if they do not own a mobile phone) requesting the results of a late afternoon USG measurement (see below), as well as on the schedule of telephone calls (see below).

5. *Scheduled telephone calls and follow-up study visits.* Participants in the intervention group will be contacted by telephone (at weeks 1, 3 and 6, then monthly in year 1, and 3-monthly for years 2–3) and reviewed face-to-face at all study visits by the Study Dietitian to assess compliance with fluid intake, discuss USG results, ensure that blood and urine samples are collected and to record any AEs and new medications commenced. If necessary, the fluid prescription will be adjusted depending on the results of progress 24 hours urine osmolality and USG. The study dietitian will monitor compliance with protein and sodium recommendations using 24 hours recalls at all study visits.

6. *Response to SMS text messages.* To provide a quantitative measure of adherence, participants in the intervention group will be required to reply (within 12 hours) to an SMS message requesting the results of a late afternoon USG. The SMS message will be delivered randomly twice a week in months 0–6, then monthly for the duration of the study.

7. *Rationale for the intervention.* Previous trials in patients with recurrent nephrolithiasis in Italy and Israel have shown that a long-term increase in fluid intake can be achieved with targeted education provided at clinic visits alone[33 34] but, for the current study in Australia, telephone coaching by a dietitian and the above tools are included to boost compliance and continued participation.[35] A systematic review revealed that self-monitoring of USG is a critical tool to enhance the implementation of increased fluid intake in clinical trials.[36] SMS texting is a method preferred by consumers in health interventions.[37]

## Standard treatment

Both study groups will continue to receive the current standard of care for ADPKD as specified by the treating nephrologist. Hypertension will be treated with ACE inhibitors or angiotensin receptor blockers as first-line agents. If hypertension remains inadequately controlled, use of additional antihypertensive agents will be at the discretion of the treating nephrologist, but treatment with diuretics will be contraindicated. The frequency of study visits in the trial is similar to standard nephrological care in ADPKD with CKD stages 1–3.

## Primary end point

The primary end point is the percentage annual change in height-adjusted total kidney volume (ht-TKV) from baseline to 36 months as determined by serial MRI. Dialysis-dependent kidney failure takes decades to develop in ADPKD, making it impractical to use serial changes in serum creatinine and eGFR for assessing treatment efficacy. Ht-TKV is a highly sensitive surrogate marker that measures exponential cyst growth, the key parameter of ADPKD progression and has been used as a primary end point in many pivotal clinical trials. The longitudinal Consortium for Radiologic Imaging Studies of PKD (CRISP-I) study established that the rate of increase in TKV is relatively constant (~5.6%/year) and can be quantified with high precision by MRI (reliability coefficient, 0.998; mean coefficient of variation, 0.01%).[2] In CRISP-II, adjusting for height reduced variability in TKV, a baseline ht-TKV value of >600 mL/m predicted development of CKD stage 3 over an 8-year period (area under the curve=0.84; sensitivity, 74%; specificity, 75%).[38] The magnitude of the rate of increase in ht-TKV predicted the risk of ESKD.

## Secondary end points

1. *Systemic AVP activity.* Markers will be serum copeptin,[5 24] 24 hours urine osmolality and volume.
2. *Kidney disease progression.* Markers will be rate of eGFR decline (0 and 3 to 36+ months); resting mean arterial pressure; urine albumin-to-creatinine ratio and kidney pain as assessed by the HALT-PKD questionnaire.
3. *Treatment adherence.* Measured by proportion of the intervention group responding within 12 hours to SMS texts requesting results of the late afternoon USG; proportion of the intervention group with 24 hours urine osmolality ≤270 mOsmol/L.
4. *Safety end points.* Measured by proportion of participants with serum $Na^+$ <130 mmol/L; episodes of serious AEs (SAEs).
5. *Patient acceptability.* Measured by a Treatment Acceptability questionnaire[39] and the proportion of participants withdrawing from the study.

6. *Quality of life.* Measured by the Kidney Disease Quality of Life short form (KDQOL-SF) 1.3 tool.
7. *Healthcare utilisation.* At each study visit, participants are asked if have had any new diagnosis, hospital visits, seen a general practitioner or changed medication since the previous visit.

Study end points were developed to ensure that appropriate health economic analyses can be undertaken at the conclusion of the trial. Future linkage with the Australian and New Zealand Dialysis and Transplant Registry will enable long-term outcomes (eg, time to reach ESKD) to be determined with minimal cost.

## Study measurements

1. *Ht-TKV* (Table 2). Renal MRI will be performed three times (at baseline and at months 18 and 36) during the study to assess the annual rate of change in ht-TKV in the study groups. MRIs will be performed in the radiology departments of the respective study centres or an external facility (approved by the investigators). A standardised protocol for image acquisition will be used,[10] and the MRI protocol and image quality will be validated in test images.[10] The MRI scans will be de-identified and encrypted, then analysed at the Translational PKD Centre at Mayo Clinic, Rochester, Minnesota, USA by blinded personnel to quantify TKV. Baseline TKV for each patient will be determined by performing kidney segmentation semi-automatically on the T2-weighted MRI using the MIROS software package. This algorithm outputs a complete segmentation after the user quickly defines crude polygon contours of each kidney every third slice.[40] The interactive toolkit included in the package is then used to perform quality assurance and finalise the segmentation on each baseline image. Thereafter, TKV will be measured in all follow-up T2-weighted MRI scans using an automated registration-based segmentation technique, as validated previously.[41] A final quality control check will also be performed on these follow-up scans using the interactive toolkit. Finally, fractional cyst volume will be calculated using an automated cyst segmentation technique along with a final quality check.

2. *Serum and urine electrolytes, creatinine and osmolality.* Outpatient serum and 24 hours urine will be collected at baseline and at months 3, 6, 9, 12, 18, 24, 30 and 36 for measuring electrolytes, creatinine, urea, urate and osmolality. In participants from the intervention group, additional samples will be collected at weeks 3 and 6 for an initial safety check (for hyponatraemia) and for titration of the water prescription. Following the last treatment visit (month 36), blood tests will be performed 2 and 4 weeks later and a 24-hour urine collection will be performed 4 weeks later, in all groups.

3. *Systemic AVP activity.* Venous blood will be collected at all study visits to measure serum copeptin (a pre-pro hormone that is a stable biomarker for AVP). Longitudinal studies show that serum copeptin is a determinant of TKV and eGFR decline in ADPKD[42 43] consistent with the hypothesis that AVP mediates renal cyst growth. Serum samples will be frozen at −80°C, and copeptin will be measured in batches using a sandwich immunoluminometric assay (CT-proAVP Kryptor, B.R.A.H.M.S, Thermo Fisher Scientific, Germany).

## Measures to reduce bias

The trial intervention is an unblinded behavioural modification. Bias and contamination could be introduced by a participant's expectations and prior knowledge of the hypothesised role of fluid intake in ADPKD. To minimise the effect of this problem:

► Both groups will be educated about the fact that fluid requirement in ADPKD is not known and that the intervention may be equally beneficial. The patient information consent form has been written in an objective, neutral manner to accurately reflect current evidence, and does not discuss the hypothesised benefits of fluid intake in ADPKD.
► Appointments will be scheduled for the two study groups at different times of the day to minimise the chance of control group participants meeting the intervention group in the waiting room.
► Outcome assessors will be blinded to participants' treatment allocations.

## Sample size calculation

In longitudinal data from the CRISP-II cohort, the average rate of increase in ht-TKV was 5.5% per year (SD, 3.8% per year; n=201; 8-year follow-up). Preclinical and clinical studies using pharmacological inhibition or adequate hydration to inhibit ADH-mediated cyst growth resulted in similar treatment efficacy of ~50% lesser increase in kidney volume.[10] In this trial, a more modest (but still clinically important) treatment effect of 35% is nominated. Using these assumptions, a total sample size of 150 will have 87% power to detect a difference in ht-TKV of 1.9% per year, using a two-sided test and a 0.05 level of significance. Taking into account the possibility of 15% dropouts, our aim is to include 180 participants (n=90 per arm).

The estimation of dropouts was based on experience from previous clinical trials in ADPKD. Furthermore, it was suspected that there would be significant interest for participants to remain in this study due to the low risk of AEs with the intervention; as well as the strong interest and motivation expressed by patients with PKD at the study centres, in part due to genetic nature of the disease and the paucity of opportunities for clinical research in PKD in the past. In reality, however, the exact proportion of subjects that dropout and/or withdraw from the intervention (protocol deviation) will not be known until the trial has concluded, and is a secondary outcome measure to assess the efficacy of the intervention.

**Table 2**  Schedule of assessments

Column groupings: "Weeks" spans columns 0–160; "Post-treatment period" spans columns 156, 158 and 160.

| | Screen | 0 | 1 | 3 | 6 | 12 | 26 | 36 | 52 | 64 | 78 | 90 | 104 | 116 | 130 | 142 | 156 | 158 | 160 |
|---|---|---|---|---|---|---|---|---|---|---|---|---|---|---|---|---|---|---|---|
| **Study visits** | 1 | 2 | | | | | 3 | | 4 | | 5 | | 6 | | 7 | | 8 | | 9 |
| **Visits to external pathology collection centre*** | 1 | 2 | | 3† | 4† | 5 | 6 | 7 | 8 | | 9 | | 10 | | 11 | | 12 | 13 | 14 |
| **Scheduled telephone call from trial staff (group B)‡** | | | X | X | X | X | X | X | X | X | X | X | X | X | X | X | X | X | X |
| **Procedures and evaluations** | | | | | | | | | | | | | | | | | | | |
| Written informed consent | X | | | | | | | | | | | | | | | | | | |
| Inclusion and exclusion criteria | X | X | | | | | | | | | | | | | | | | | |
| Randomisation§ | | X | | | | | | | | | | | | | | | | | |
| Demographics (age, race, gender, education level and health insurance status) | X | | | | | | | | | | | | | | | | | | |
| Medical/ADPKD history | X | | | | | | | | | | | | | | | | | | |
| Concomitant medications | X | X | X | X | X | X | X | X | X | X | X | X | X | X | X | X | X | X | X |
| AEs | | X | X | X | X | X | X | X | X | X | X | X | X | X | X | X | X | X | X |
| Kidney pain assessment | X | | | | | | X | | X | | X | | X | | X | | X | | X |
| Quality of life assessment | | X | | | | | X | | X | | X | | X | | X | | X | | |
| Qualitative evaluation (group B) | | | | | | | X | | X | | | | X | | | | X | | |
| Patient acceptability question (group B) | | | | | | | X | | X | | | | X | | | | X | | |
| Dietary and fluid intake¶ | X | X | | | | | X | | X | | X | | X | | X | | X | | X |
| Vital signs (heart rate, abdominal girth), weight | X | X | | | | | X | | X | | X | | X | | X | | X | | X |
| Height | X | | | | | | | | | | | | | | | | | | |
| Office BP | X | X | | | | | X | | X | | X | | X | | X | | X | | X |
| Physical examination | X | | | | | | | | | | | | | | | | | | |
| Urinalysis (dipstick) | X | X | | | | | X | | X | | X | | X | | X | | X | | X |
| TKV (by MRI) | X | | | | | | | | | | X | | | | | | X | | |
| 24 hours urine osmolality (external pathology centre) | X** | | | | | X†† | X‡‡ | X†† | X‡‡ | | X‡‡ | | X‡‡ | | X‡‡ | | X** | | |
| Routine blood tests (external pathology centre) | X | X | | X | X | X | X | X | X | | X | | X | | X | | X | X | X |

Continued

**Table 2** Continued

| Study visits | Screen | Weeks | | | | | | | | | | | | | | | Post-treatment period | | |
|---|---|---|---|---|---|---|---|---|---|---|---|---|---|---|---|---|---|---|---|
| | | 0 | 1 | 3 | 6 | 12 | 26 | 36 | 52 | 64 | 78 | 90 | 104 | 116 | 130 | 142 | 156 | 158 | 160 |
| Study visits | 1 | 2 | | | | 3 | | | 4 | | 5 | | 6 | | 7 | | 8 | | 9 |
| Visits to external pathology collection centre* | 1 | 2 | | 3† | 4† | 5 | 6 | 7 | 8 | | 9 | | 10 | | 11 | | 12 | 13 | 14 |
| Scheduled telephone call from trial staff (group B)‡ | | | X | X | X | X | X | X | X | X | X | X | X | X | | X | X | X | X |
| Serum copeptin | X | X | | | | | X | X | X | | X | | X | | X | | X | | X |
| DNA for PKD gene analysis (blood sampling)§§ | X | X | | | | | | | | | | | | | | | | | |
| Urine and blood spot samples (biomarkers)¶¶ | X | X | | | | | X | | X | | X | | X | | X | | X | | X |

*The patients are to visit an external pathology collection centre for blood and urine sample collection.

†Group B patients only for safety check and titration of water prescription.

‡Patients in group B will receive scheduled telephone calls from the study nurse (with the dietitian, as needed) to review changes in health, fluid intake, USG, laboratory results, AE and provided coaching to alter water intake (if needed). These calls will be made at week 1, 3, 6 and then monthly in year 1 and 3-monthly in years 2–3 (and as required). Patients in group B will only be contacted if the pathology results meet the criteria for an AE.

§Patients randomised to group B will receive specific advice on the amount of fluid required per day to reduce the urine osmolality to ≤270 mOsmol/kg. Patients in group B will be asked to self-monitor their USG regularly at home using urine dipstick (provided to the patients), and adherence to the intervention will be quantified by patient responses to SMS text, sent twice weekly during months 0–6 and then monthly for the duration of the study, asking for the results of that day's USG.

¶All patients will conduct two 24 hours urine collections and record their fluid intake for those 2 days prior to baseline visit. The study dietitian will provide group B patients with diet and fluid assessments and review during all study visits to ensure prescribed fluid intake is achieved and to validate a fluid intake tool.

**To be collected following the screening visit and prior to the final visit (two collections each).

††To be collected between study visits (one collection only).

‡‡To be collected prior to the study visit (one collection only).

§§To be collected at screening for PKD gene analysis.

¶¶To be collected at all study visits for exploratory renal biomarkers (stored at −80°C).

ADPKD, autosomal dominant polycystic kidney disease; AE, adverse event; BP, blood pressure; PKD, polycystic kidney disease; SMS, short message service; USG, urine-specific gravity.

## Statistical methods and data management

Intention-to-treat principles will be followed. Patient characteristics (age, sex, ht-TKV, eGFR) will be compared at baseline, and analysis of covariance will be used to analyse $\log_{10}$ ht-TKV at months 18 and 36, with baseline $\log_{10}$ ht-TKV as the covariate. Assumptions of the analysis of covariance of normally distributed residuals and constant variance will be assessed with normal probability plots and residual versus fitted plots. Linear mixed-effects models will be used to test the interaction between treatment groups and time, if assumptions of equal covariance between times cannot be guaranteed. The adherence distribution will be constructed in the intervention arm by using the proportion of times a subject responds to the SMS. Outcome measure will be transformed to appropriate normality if required, and the $X^2$ test or Fisher's exact test will be used to test the association between categorical variables. The data management plan and the electronic case report forms have been developed in the OpenClinica platform, harmonised using National Institutes of Health-Clinical Data Interchange Standards Consortium (NIH-CDISC) terminology for ADPKD to enable future data sharing.[44] The Research Data Storage Plan has been approved by University of Sydney.

## Process evaluation of the intervention

To understand patient experiences and attitudes about the intervention (prescribed water intake), a process evaluation substudy will be conducted, using methods previously described.[45] Measures assessed will include the frequency, timing and difficulties experienced with intervention and the intervention tools (telephone coaching, USG, SMS, water bottles, water and diet guidebook, website). The data will be collected using a questionnaire during study visits, and a semi-structured telephone interview will be conducted in a minimum of 30 intervention group participants following the final study visit. Thematic analysis will identify key facilitators and barriers to intervention uptake.

## Economic evaluation

A trial-based economic evaluation, from the perspective of the health funder will be conducted. Information on quality-adjusted life years (QALYs) will be collected using the short form 6D calculated from the KDQOL (SF-1.3). Self-reported healthcare utilisation and costs will be collected at routine clinic visits and intervention costs (staff, training, capital costs and consumables) will also be included. Using the mean costs and the mean health outcomes in each trial arm, the incremental costs per QALY of the intervention group compared with the control group will be calculated; results will be plotted on a cost-effectiveness plane. Bootstrapping will be used to estimate a distribution around costs and health outcomes, and to calculate the CIs around the incremental cost-effectiveness ratios. One-way and multiway sensitivity analyses will be conducted around key variables. A cost-effectiveness acceptability curve will be plotted to provide information about the probability that the intervention is cost-effective, given willingness to pay for each additional QALY gained.

## Comparison of prescribed water intake with pharmacological suppression of AVP

It would be important for consumers, policy-makers and other stakeholders to be informed of the direct comparative treatment effects of suppressing the AVP pathway in ADPKD using prescribed water intake or by pharmacological inhibition with a vasopressin receptor antagonist. In this regard, the addition of a third trial arm to compare prescribed water intake with open-label treatment with a vasopressin type 2 receptor antagonist (such as tolvaptan, Otsuka Pharmaceuticals; or Lixivaptan, Palladio Biosciences) in the PREVENT-ADPKD study was considered. However, the cost of study drug was prohibitive for government grant funding (~$A15 million assuming treatment with tolvaptan at up to 90/30 mg dosage for n=90 participants for 3 years), and commercial sponsorship has not been achieved. Therefore, the investigators will conduct a modelled economic evaluation versus tolvaptan, using trial costs and outcomes, supplemented with best available published evidence to consider costs and outcomes over a longer time horizon to account for future benefits in terms of delayed commencement of dialysis, quality of life and life expectancy. One-way and multiway sensitivity analyses will be conducted around key variables and a probabilistic sensitivity analysis will estimate uncertainty in all parameters.

## Data and safety monitoring board

Prescribed fluid intake is considered a safe treatment. The risk of an SAE related to hyponatraemia is expected to be very low in the study population (<1:100 000), however, to mitigate this, serum sodium concentrations will be monitored regularly throughout the trial. An independent data monitoring safety board (DSMB) has been appointed to monitor the safety and conduct of this trial. Specific aspects that will be reviewed include recruitment rate and losses to follow-up, data quality, compliance with the protocol by participants and investigators, evidence for treatment harm (treatment group differences in SAEs), protocol modifications and continuing appropriates of participant information. The DSMB Charter was ratified in October 2016 and the first DSMB meeting occurred in December 2016.

## Trial monitoring

The monitoring of the trial will be performed independently by the Australasian Kidney Trials Network using a combination of remote-monitoring tools and site visits.

## Study recruitment, retention and study limitations

As discussed earlier, to achieve the target recruitment and maintain retention, multiple recruitment strategies including presentations to local nephrologists, media advertisement and engagement of multiple sites for

**Table 3** Proposed timeline of the PREVENT-ADPKD trial

| Year | Milestone (*italics denote milestone has been completed*) |
|---|---|
| 2012–2015 | *Study protocol developed (V.4); Australian and New Zealand Clinical Trial registration; lead-site ethics committee approval; trial endorsed by AKTN; data and biostatistical management plan developed; DSMB appointed; randomisation list and electronic case report forms finalised; recruitment at Westmead Hospital started; intervention group supporting tools developed* |
| 2016 | *Commenced recruitment at Norwest Private Hospital, Nepean Hospital and Gosford Renal Research*<br>*50% planned recruitment completed* |
| 2017 | *Commence recruitment at other Australian sites*<br>*75% recruitment completed in July 2017*<br>Plan to complete recruitment at the end of 2017 |
| 2018–2019 | Follow-up of study participants |
| 2020–2021 | Last participant follow-up<br>Study close-out, data analysis; report key findings |

AKTN, Australasian Kidney Trials Network; DSMB, Data and Safety Monitoring Board.

participants' convenience to attend study visits will be performed. The individualised nature of the study treatment and regular direct participant contact are key to the retention of participants.[46] There are no competing studies in ADPKD in the region of the study centres, and regulatory approval of tolvaptan for use in ADPKD has been attained in Europe, Canada and Japan, but not in Australia and the USA, and therefore will not affect recruitment.[47]

The study intervention (prescribed fluid intake) is reliant on the behavioural change to drinking habits of the participant to reach 24 hours urine osmolality ≤270 mOsmol/L. The tailored intervention with a variety of supporting tools as described under *study intervention (group B)* adopts a similar model to other successful long-term behaviour change interventions.[37 48] However, due to the nature of the western diet, adherence to the trial intervention (both prescribed water intake and limitation of dietary salt and protein restriction) can be difficult even with intensive dietary counselling. Progress results from 24 hours urine volume, osmolality and sodium will be monitored to assess compliance of group B participants to the trial intervention. Additionally, bias and contamination could be introduced by the patient's expectations and prior knowledge of the hypothesised role of fluid intake in ADPKD, as discussed earlier. Finally, another limitation of the study is that renal function will be estimated by the CKD-EPI equation and it is known that this may not reliably predict longitudinal changes in patients with ADPKD,[49] which is one of the principal reasons why TKV has been used as the primary outcome measure.

### Proposed timeline and current status of the trial

The study has been in the development phase from 2012 to 2015, and trial recruitment commenced on 9 December 2015 (see table 3). In 2016, the model of a mobile research team was implemented and the active study centres included Westmead, Nepean and Norwest Private Hospitals in Western Sydney and

Gosford Renal Research on the NSW Central Coast. In 2017, additional study centres commenced, and as of July 2017, 75% of the intended recruitment has been completed and 100% recruitment is anticipated by the end of 2017.

### Outcomes and significance

This trial will determine if fluid intake prescribed to maintain isotonic urine (implemented by coaching, SMS text reminders and self-monitoring of USG by dipstick) reduces the progression of TKV in CKD stages 1–3 due to ADPKD. While a negative result of a properly performed study will be significant in that it will settle the controversy regarding fluid intake in ADPKD, a positive study result will provide an inexpensive, widely generalisable and safe approach to slow renal cyst growth, and one that could be easily taken up in clinical practice and well-tolerated by consumers.[50] In the best-case scenario, if prescribed fluid intake is found to reduce the annualised rate of increase in TKV by 50%, the development of ESKD could be delayed by 6.5 years and life expectancy extended by 2.6 years,[51] at a negligible cost over standard treatment, but resulting in considerable cost savings for future treatments of ESKD. Even at lower efficacy this treatment option will be extremely good value for money and this is of vital importance in low-income countries where access to novel drugs and chronic dialysis are restricted due to lack of affordability and availability. However, if the hypothesis is proven, the largest impact will be in children and at-risk individuals with ADPKD (where its introduction in early life could potentially prevent the onset of ESKD).

### Ethics and dissemination

The results will be submitted to national and international conferences and peer-reviewed medical journals for consideration of publication, after the last participant has completed the final study visit and/or in the event of early termination of the trial for any reason.

**Author affiliations**
$^{1}$Centre for Transplant and Renal Research, The Westmead Institute for Medical Research, The University of Sydney, Sydney, Australia
$^{2}$Department of Renal Medicine, Westmead Hospital, Western Sydney Local Health District, Sydney, Australia
$^{3}$The Kidney Institute, Division of Nephrology and Hypertension, Kansas University Medical Center, Kansas City, Kansas, USA
$^{4}$School of Life and Environmental Sciences, The Charles Perkins Centre, The University of Sydney, Sydney, Australia
$^{5}$Department of Renal Medicine, St. George Hospital, Sydney, Australia
$^{6}$Department of Renal Medicine, Sir Charles Gairdner Hospital, Nedlands and the Harry Perkins Institute of Medical Research, University of Western Australia, Sydney, Australia
$^{7}$Research and Education Network, Westmead Hospital, Western Sydney Local Health District, Sydney, Australia
$^{8}$McCloud Consulting Group, Gordon, Australia
$^{9}$North Shore Nephrology, Crows Nest, Australia
$^{10}$Translational Polycystic Kidney Disease Center, Mayo Clinic, Rochester, Minnesota, USA
$^{11}$Department of Renal Medicine, Prince of Wales Hospital, Eastern Sydney Health District and the University of New South Wales, Randwick, Australia
$^{12}$Department of Radiology, Westmead Hospital, Western Sydney Local Health District, Sydney, Australia
$^{13}$Faculty of Health Sciences, The University of Sydney, Sydney, Australia
$^{14}$Department of Renal Medicine, Bathurst Base Hospital, Bathurst, Australia
$^{15}$Australasian Kidney Trials Network, University of Queensland at Princess Alexandra Hospital, Woolloongabba, Australia
$^{16}$School of Public Health, University of Sydney, Sydney, Australia
$^{17}$Department of Renal Medicine, Canberra Hospital, Garran, Australia
$^{18}$Department of Immunology and Infectious Diseases, John Curtin School of Medical Research, Australian National University, Canberra, Australia
$^{19}$Gosford Nephrology, Gosford, Australia
$^{20}$Department of Renal Medicine, Norwest Private Hospital, Sydney, Australia
$^{21}$Department of Renal Medicine, Wollongong Hospital, Illawarra Shoalhaven Local Health District, Wollongong, Australia
$^{22}$Department of Renal Medicine, Liverpool Hospital, Southwestern Sydney Local Health District, Liverpool, Australia
$^{23}$Renal Research, Gosford, Australia
$^{24}$Department of Renal Medicine, Nepean Hospital, Nepean Blue Mountains Local Health District, Sydney, Australia
$^{25}$Nepean Clinical School, The University of Sydney Medical School, Sydney, Australia
$^{26}$Department of Nephrology, John Hunter Hospital, Newcastle, Australia

**Acknowledgements** The study protocol was reviewed by the Scientific Committee of the AKTN and the NHMRC Grant Review Panels (2013, 2014, 2015, 2016) who provided feedback on the study design. The authors thank Dr William Clark and Dr Louise Moist (Division of Nephrology, London Health Sciences Centre, Canada) and Dr Hakam Gharbi (Danone Nutricia) for helpful discussions on the study protocol.

**Contributors** All authors contributed and developed the study protocol. The rationale and hypothesis for the trial arose from reviews and pilot studies in humans authored by JGG and supported by the results of the TEMPO 251 clinical trial. GKR developed the initial version of the study protocol with JJG, DCHH, VWL and KS. ATYW, CM, MAF and AR contributed to the implementation of the intervention. KB provided initial biostatistical advice and suggested using SMS texting as a potential measure of compliance in the control group. JH and PM provided biostatistical advice and developed the biostatistical plan. JC developed eCRF and data management protocols. KH and MH developed the protocol for health economic analysis. SF, TP, VT, BJE, TLK and MEE developed protocols on analysis of TKV by MRI. DWJ and CHH provided input on trial oversight and overall management. SC, SVB, NB, IH, SHJ, JM, CM, AP, SDR, EV provided additional input into the study protocol.

**Funding** The development and commencement of the trial was funded by University of Sydney Bridging Grants (2014, 2016), Westmead Medical Research Foundation, the Western Sydney Local Health District and an investigator-initiated research grant from Danone Nutricia Research (France).

**Competing interests** The study protocol was developed independently by the authors listed of this manuscript. GR is the sole principal investigator listed on the grant received from Danone Nutricia Research to conduct this trial. The grant was awarded in December 2015 and is being administered by the study sponsor (Western Sydney Local Health District). GR has received travel support from Danone Nutricia Research to attend an international meeting on hydration (2016 and 2017). No other authors have competing interests to declare.

**Patient consent** Not required.

**Ethics approval** The trial was approved by the Human Research Ethics Committee of the Western Sydney Local Health District in 2014.

**Provenance and peer review** Not commissioned; externally peer reviewed.

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
