## [Reviewer comments · BMJ Open]

ARTICLE DETAILS

TITLE (PROVISIONAL)	The study protocol of a randomised controlled trial to determine efficacy and safety of prescribed water intake to prevent kidney failure due to autosomal dominant polycystic kidney disease (PREVENT-ADPKD)
AUTHORS	Wong, Annette; Mannix, Carly; Grantham, Jared; Allman-Farinelli, Margaret; Badve, Sunil; Boudville, Neil; Byth, Karen; Chan, Jessie; Coulshed, Susan; Edwards, Marie; Erickson, Bradley; Fernando, Mangalee; Foster, Sheryl; Haloob, Imad; Harris, David; Hawley, Carmel; Hill, Julie; Howard, Kirsten; Howell, Martin; Jiang, Simon; Johnson, David; Kline, Timothy; Kumar, Karthik; Lee, Vincent; Lonergan, Maureen; Mai, Jun; McCloud, Philip; Peduto, Anthony; Rangan, Anna; Roger, Simon; Sud, Kamal; Torres, Vincent; Vliayuri, Eswari; Rangan, Gopala

VERSION 1 – REVIEW

REVIEWER	Giuseppe Remuzzi IRCCS - Istituto di Ricerche Farmacologiche Mario Negri. Bergamo, Italy
REVIEW RETURNED	04-Aug-2017

GENERAL COMMENTS	Wong and colleagues present the protocol of a multicenter, prospective, open-label, randomized clinical trial investigating the long-term efficacy and safety of prescribed water intake in slowing the rate of autosomal dominant polycystic kidney disease (ADPKD) progression. In this study 180 ADPKD patients with estimating glomerular filtration rate (eGFR) ≥ 30 mL/min/1.73 m² will be randomly allocated to continue their usual water consumption or to increase water intake in order to reduce urine osmolality to ≤ 270 mosmol/kg for the next 36 months, both on top of standard treatment. The primary endpoint will be the percentage annual change in height-adjusted total kidney volume from baseline to 36 months. The Authors may wish to consider the following drawbacks: 1. The aim of present study is to compare the long-term efficacy and safety of high water intake compared to usual water consumption on the rate of total kidney volume increase in patients with ADPKD. An economic evaluation versus tolvaptan will be carried out using trial costs and outcomes (Page 25, lines 18-22). However, it would have been much more valuable to assess the efficacy and safety of high water intake compared to tolvaptan on ADPKD progression in the setting of a randomized clinical trial. Indeed, since the benefits of tolvaptan on ADPKD (i.e., slowing the
---

progressive increase in total kidney volume and the decline in renal function) have been ascribed to inhibition of V2-receptor activation and the eventual suppression of cyclic adenosine monophosphate (N Eng J Med 2012; 367:2407-18), it is reasonable to expect that a similar favourable effect on the course of ADPKD could be achieved with high fluid intake alone, because this also suppresses vasopressin release and cyclic adenosine monophosphate formation. Actually, considering the adverse effects (in particular hepatic toxicity) and the extremely high cost of tolvaptan, high water intake would be a safer and far cheaper alternative, provided that they are equally effective in slowing ADPKD progression.

2. A limitation of this study is that the sample size was calculated accounting for a drop-out rate of only 15%. The expectation that most of the patients assigned to the intervention group will be actually compliant to a high water intake regimen (about 3-4 L per day) for as long as 36 months sounds quite optimistic. Indeed, in a previous work investigating the potential benefits of water loading on renal graft function preservation in kidney transplant recipients, patients' adherence to the prescribed high fluid intake (4L per day) abated over the 12 month study period (J Ren Nutr 2011; 21:499-505). Moreover, in a pilot study in ADPKD patients, in three of the eight patients (i.e., 37.5%) urinary osmolality could not be adequately suppressed during one week of oral water loading (Clin J Am Soc Nephrol 2011; 6:192-197). The rationale for the anticipated 15% drop-out rate should be explained.

3. At screening visit venous blood samples were collected for DNA analysis (Page 10, lines 11-12). Considering that mutations in the gene coding for polycystin 1 have been associated to more rapid renal disease progression compared to mutation in polycystin 2, I wonder whether ADPKD patients were randomized to usual water consumption or increased water intake according to either PKD1 or PKD2 mutations. A comment on this issue is welcome.

4. Since a high dietary solute load (due to high salt and/or protein intakes) requires a higher fluid intake to maintain urine dilution, the study dietician will provide dietary advice to limit sodium intake to 80-100 mmol/day (i.e., 1.8-2.3 mg/day) in patients randomized to increase water intake (Page 13, lines 1-2). However, the nature of the Western diet and the excess of salt in processed food mean that a careful dietary plan may not be sufficient to achieve and maintain the proposed target of sodium intake in the long-term run. This issue deserves a comment. Moreover, it should be clarified whether 24 h urinary sodium excretion will be monitored to assess actual adherence to low sodium diet.

5. As a secondary end-point of this clinical trial, renal function will be estimated by means of the Chronic Kidney Disease Epidemiology Collaboration (CKD-EPI) equation (Page 16, lines 4-5). However, creatinine-based prediction formulas, including CKD-EPI, have been shown to unreliably estimate measured GFR, and failed to predict GFR changes over time in a cohort of adult patients with ADPKD, independent of their baseline kidney function (PLoS One 2012; 7:e32533). This study limitation should be acknowledged.

6. To avoid that bias could be introduced by prior knowledge of the hypothesised role of fluid intake in ADPKD, patients will be educated about the notion that fluid requirement in ADPKD is not known (Page 23, lines 1-7). However, a previous study evaluating the effects of

	increased water intake on ADPKD progression showed that even before randomisation several patients had already been informed on the potential favourable effect of high water intake on the course of their disease (Nephrol Dial Transplant 2014; 29:1710-1719). Thus, it is reasonable to expect that in the present study patients randomized to the usual water intake, but already informed regarding the potential benefits of fluid intake on disease course, will be motivated to drink large amounts of water, eventually reducing the difference in urine osmolality between the two study groups and masking any possible benefit(s) of water intake on ADPKD progression. A comment on this issue is welcome. Minor points:  - In the Introduction it was stated that in ADPKD renal cysts derive from epithelial cells of the distal tubule and collecting duct (Page 5, lines 8-9; Page 5, lines 21-22). However, kidney cysts can also originate from other segments of the nephron, such as proximal tubule (Nat Clin Pract Nephrol 2006; 2: 40-55). These sentences should be revised accordingly. - Based on the Legend of Figure 1, the schema of the PREVENT-ADPKD Trial Design has been adapted from reference #9. However, the cited reference (Nat Med 2003; 9:1323-26) deals with the use of a vasopressin V2-receptor antagonist in mice with a rapidly progressive form of polycystic kidney disease. Proper reference(s) should be quoted.
--	---

REVIEWER	Adam E. Mikołajczyk, MD Transplant Hepatology Fellow University of Chicago Medicine Chicago, IL USA
REVIEW RETURNED	14-Aug-2017

GENERAL COMMENTS	Rangan and colleagues have crafted a well-designed protocol to study the effects of prescribed water intake on the progression of renal disease in ADPKD. This manuscript does a nice job of providing a detailed description of the study design. A few comments that should be addressed prior to publication:  1. One of the exclusion criteria listed is "risk of non-compliance with trial procedures". Is this a subjective assessment alone or were some objective measures (e.g. number of no shows to clinic, etc.) considered. 2. "Concomitant conditions or treatments likely to confound endpoint assessments" was also listed as an exclusion criteria. But it is important to list what such entities would preclude involvement in the trial. If there was no pre-defined list of conditions/treatments, it seems that this a potential area of bias for those individuals enrolling patients. 3. This study utilizes standardized quality of life questionnaires, but are metrics specific to side effects of the intervention, such as quality of sleep (because of presumed increase in rate of nocturia) and level of satisfaction with an increased frequency of urination (which may be disruptive when at work or running errands), going to be measured? If so, are they included in the secondary endpoint
---

	"patient acceptability"? If not, it still would be helpful to provide a copy of this questionnaire as a supplement to know what items are being assessed. 4. Are the USG assessments randomly occurring or are they scheduled in a predictable manner? This should be clarified. It would be much more powerful to have these assessments occur randomly after the initial two weeks, because otherwise patients may only increase fluid intake on the days that they know their urine will be assessed. If randomly occurring, patients will be more likely to adhere to the prescribed water intake on a daily basis because they will never know when they are going to be assessed. 5. Page 17, line 49, "and a baseline ht-TKV vale of > 600"-- the "and" should be removed
--	--

VERSION 1 – AUTHOR RESPONSE

Reviewer #1

Comment #1: The aim of present study is to compare the long-term efficacy and safety of high water intake compared to usual water consumption on the rate of total kidney volume increase in patients with ADPKD. An economic evaluation versus tolvaptan will be carried out using trial costs and outcomes (Page 25, lines 18-22). However, it would have been much more valuable to assess the efficacy and safety of high water intake compared to tolvaptan on ADPKD progression in the setting of a randomized clinical trial. Indeed, since the benefits of tolvaptan on ADPKD (i.e., slowing the progressive increase in total kidney volume and the decline in renal function) have been ascribed to inhibition of V2-receptor activation and the eventual suppression of cyclic adenosine monophosphate (N Eng J Med 2012; 367U2407-18), it is reasonable to expect that a similar favourable effect on the course of ADPKD could be achieved with high fluid intake alone, because this also suppresses vasopressin release and cyclic adenosine monophosphate formation. Actually, considering the adverse effects (in particular hepatic toxicity) and the extremely high cost of tolvaptan, high water intake would be a safer and far cheaper alternative, provided that they are equally effective in slowing ADPKD progression.

Authors' Response: We agree with Reviewer #1. The manuscript has been revised to include an extra paragraph to discuss this comment. Please see page 29, paragraph 2 of the Tracked version.

Comment #2: A limitation of this study is that the sample size was calculated accounting for a drop-out rate of only 15%. The expectation that most of the patients assigned to the intervention group will be actually compliant to a high water intake regimen (about 3-4 L per day) for as long as 36 months sounds quite optimistic. Indeed, in a previous work investigating the potential benefits of water loading on renal graft function preservation in kidney transplant recipients, patients' adherence to the prescribed high fluid intake (4L per day) abated over the 12 month study period (J Ren Nutr 2011; 21U499-505). Moreover, in a pilot study in ADPKD patients, in three of the eight patients (i.e., 37.5%) urinary osmolality could not be adequately suppressed during one week of oral water loading (Clin J Am Soc Nephrol 2011; 6U192-197). The rationale for the anticipated 15% drop-out rate should be explained.

Authors' Response: The estimation of dropouts was based on experience from previous clinical trials in ADPKD. Furthermore, it was suspected that there would be significant interest for participants to remain in this study due to the low risk of adverse events with the intervention; as well as the strong interest and motivation expressed by PKD patients at the study centres, in part due to genetic nature

of the disease and the paucity of opportunities for clinical research in PKD in the past. In reality, however, the exact proportion of subjects that dropout and/or withdraw from the intervention (protocol deviation) will not be known until the trial has concluded, and is a secondary outcome measure to assess the intervention's efficacy. The manuscript has been revised to include an extra paragraph to discuss this comment. Please see page 27, paragraph 1 of the Tracked version.

Comment #3: At screening visit venous blood samples were collected for DNA analysis (Page 10, lines 11-12). Considering that mutations in the gene coding for polycystin 1 have been associated to more rapid renal disease progression compared to mutation in polycystin 2, I wonder whether ADPKD patients were randomized to usual water consumption or increased water intake according to either PKD1 or PKD2 mutations. A comment on this issue is welcome.

Authors' Response: Due to time and cost of genetic testing, the effect of PKD mutation type on trial outcomes will be assessed retrospectively. The manuscript has been revised to include an extra sentence to discuss this comment. Please see page 12, paragraph 1 of the Tracked version.

Comment #4: Since a high dietary solute load (due to high salt and/or protein intakes) requires a higher fluid intake to maintain urine dilution, the study dietician will provide dietary advice to limit sodium intake to 80-100 mmol/day (i.e., 1.8-2.3 mg/day) in patients randomized to increase water intake (Page 13, lines 1-2). However, the nature of the Western diet and the excess of salt in processed food mean that a careful dietary plan may not be sufficient to achieve and maintain the proposed target of sodium intake in the long-term run. This issue deserves a comment. Moreover, it should be clarified whether 24 h urinary sodium excretion will be monitored to assess actual adherence to low sodium diet.

Authors' Response: We agree with Reviewer #1. Due to the nature of the western diet, adherence to the trial intervention (both prescribed water intake and limitation of dietary salt and protein restriction) can be difficult even with intensive dietary counselling. Progress results from the 24-hour urine volume, osmolality and sodium will be monitored to assess compliance of Group B participants with the trial intervention. The manuscript has been revised to include an extra sentence to discuss this comment. Please see page 31, paragraph 1 of the Tracked version.

Comment #5: As a secondary end-point of this clinical trial, renal function will be estimated by means of the Chronic Kidney Disease Epidemiology Collaboration (CKD-EPI) equation (Page 16, lines 4-5). However, creatinine-based prediction formulas, including CKD-EPI, have been shown to unreliably estimate measured GFR, and failed to predict GFR changes over time in a cohort of adult patients with ADPKD, independent of their baseline kidney function (PLoS One 2012; 7:e32533). This study limitation should be acknowledged.

Authors' Response: We agree with Reviewer #1. Please see page 31, paragraph 1 of the Tracked version.

Comment #6: To avoid that bias could be introduced by prior knowledge of the hypothesised role of fluid intake in ADPKD, patients will be educated about the notion that fluid requirement in ADPKD is not known (Page 23, lines 1-7).

However, a previous study evaluating the effects of increased water intake on ADPKD progression showed that even before randomisation several patients had already been informed on the potential favourable effect of high water intake on the course of their disease (Nephrol Dial Transplant 2014; 29U1710-1719). Thus, it is reasonable to expect that in the present study patients randomized to the usual water intake, but already informed regarding the potential benefits of fluid intake on disease

course, will be motivated to drink large amounts of water, eventually reducing the difference in urine osmolality between the two study groups and masking any possible benefit(s) of water intake on ADPKD progression. A comment on this issue is welcome.

Authors' Response: We agree with Reviewer #1. This is discussed in the original manuscript in the section "Measure to reduce bias" (see page 26 of the tracked version) but we have also mentioned this again briefly in the section on Study Limitations (please see page 31, paragraph 1 of the Tracked version).

Comment #7: In the Introduction it was stated that in ADPKD renal cysts derive from epithelial cells of the distal tubule and collecting duct (Page 5, lines 8-9; Page 5, lines 21-22). However, kidney cysts can also originate from other segments of the nephron, such as proximal tubule (Nat Clin Pract Nephrol 2006; 2: 40-55). These sentences should be revised accordingly

Authors' Response: This sentence has been modified by removing the word "distal" from the sentences.

Comment #8: Based on the Legend of Figure 1, the schema of the PREVENT-ADPKD Trial Design has been adapted from reference #9. However, the cited reference (Nat Med 2003; 9U1323-26) deals with the use of a vasopressin V2-receptor antagonist in mice with a rapidly progressive form of polycystic kidney disease. Proper reference(s) should be quoted.

Authors' Response: The correct reference has been inserted.

Reviewer #2

Comment #1: One of the exclusion criteria listed is "risk of non-compliance with trial procedures". Is this a subjective assessment alone or were some objective measures (e.g. number of no shows to clinic, etc.) considered.

Authors' Response: We agree with Reviewer #2, and further detail is provided on the definition of non-compliance in Table 2 on page 9 of the Tracked version.

Comment #2: Concomitant conditions or treatments likely to confound endpoint assessments" was also listed as an exclusion criteria. But it is important to list what such entities would preclude involvement in the trial. If there was no pre-defined list of conditions/treatments, it seems that this a potential area of bias for those individuals enrolling patients.

Authors' Response: We agree with Reviewer #2, and further detail is provided on the definition of non-compliance in Table 2 on page 9 of the Tracked version.

Comment #3: This study utilizes standardized quality of life questionnaires, but are metrics specific to side effects of the intervention, such as quality of sleep (because of presumed increase in rate of nocturia) and level of satisfaction with an increased frequency of urination (which may be disruptive when at work or running errands), going to be measured? If so, are they included in the secondary endpoint "patient acceptability"? If not, it still would be helpful to provide a copy of this questionnaire as a supplement to know what items are being assessed.

Authors' Response: A Treatment Acceptability Questionnaire will be performed (adapted from Hunsley 1992, and this reference has been included in the revised version; please see page 19, paragraph 2 of the Tracked version). Acceptability of the trial intervention will also be assessed as part of the Process Evaluation.

Comment #4: Are the USG assessments randomly occurring or are they scheduled in a predictable manner? This should be clarified. It would be much more powerful to have these assessments occur randomly after the initial two weeks, because otherwise patients may only increase fluid intake on the days that they know their urine will be assessed. If randomly occurring, patients will be more likely to adhere to the prescribed water intake on a daily basis because they will never know when they are going to be assessed

Authors' Response: The SMSs are sent out on random days of the week. This is clarified on page 17 of the Tracked version.

Comment #5: Page 17, line 49, "and a baseline ht-TKV value of > 600"-- the "and" should be removed.

Authors' Response: This has been corrected in the revised manuscript. Please see page 18 of the tracked version.

VERSION 2 – REVIEW

REVIEWER	Giuseppe Remuzzi IRCCS - Istituto di Ricerche Farmacologiche Mario Negri, Bergamo (Italy)
REVIEW RETURNED	12-Oct-2017
GENERAL COMMENTS	In the revised version of the manuscript the Authors added the required information. In particular, the rationale for an expected drop-out rate of 15% has been provided, and the assessment of glomerular filtration rate by means of an estimation equation has been acknowledged as a limitation of the study.